# Torularhodin Alleviates Hepatic Dyslipidemia and Inflammations in High-Fat Diet-Induced Obese Mice via PPARα Signaling Pathway

**DOI:** 10.3390/molecules27196398

**Published:** 2022-09-27

**Authors:** Xingming Li, Yuliang Cheng, Jiayi Li, Chang Liu, He Qian, Genyi Zhang

**Affiliations:** 1State Key Laboratory of Food Science and Technology, School of Food Science and Technology, Jiangnan University, Wuxi 214122, China; 2Yitong Food Industry Co., Ltd., Xuzhou 221000, China; 3Collaborative Innovation Center of Food Safety and Quality Control in Jiangsu Province, Jiangnan University, Wuxi 214122, China

**Keywords:** Torularhodin, dyslipidemia, PPARα pathway, β-oxidation, inflammation, mice

## Abstract

Torularhodin is a β-carotene-like compound from *Sporidiobolus pararoseus,* and its protective effect against high-fat diet (HFD)-induced hepatic dyslipidemia and inflammation was investigated. Compared to mice of C57BL/6J fed on HFD, the addition of Torularhodin into the HFD (HFD-T) significantly reduced body weight, serum triglyceride (TG), total cholesterol (TC), low-density lipoprotein (LDL), and the inflammatory mediators of TNF-α, IL-6, IL-1β, and lipopolysaccharide (LPS). A significant increase of high-density lipoprotein cholesterol (HDL-c), which is beneficial to cholesterol clearance, was also observed in HFD-T group. Proteomic analysis showed HDL-C-c is highly correlated with proteins (e.g., CPT1A and CYP7A1) involved in lipid β-oxidation and bile acid synthesis, whereas the other phenotypic parameters (TC, TG, LDL, and inflammatory cytokines) are highly associated with proteins (e.g., SLC27A4) involved in lipid-uptake. The up-regulated anti-inflammation proteins FAS, BAX, ICAM1, OCLN, GSTP1, FAF1, LRP1, APEX1, ROCK1, MANF, STAT3, and INSR and down-regulated pro-inflammatory proteins OPTN, PTK2B, FADD, MIF, CASP3, YAP1, DNM1L, and NAMPT not only demonstrate the occurrence of HFD-induced hepatic inflammation, but also prove the anti-inflammatory property of Torularhodin. KEGG signaling pathway analysis revealed that the PPARα signaling pathway is likely fundamental to the health function of Torularhodin through up-regulating genes related to fatty acid β-oxidation, cholesterol excretion, HDL-Cc formation, and anti-inflammation. Torularhodin, as a new food resource, may act as a therapeutic agent to prevent hepatic dyslipidemia and related inflammation for improved health.

## 1. Introduction

The liver is the central player in lipid metabolism involving fatty acid metabolism, lipoprotein synthesis, and cholesterol transformation. Alterations in β-oxidation, very low-density lipoprotein secretion, and pathways involved in the metabolism of fatty acids and cholesterol could lead to excess fat buildup in the liver (steatosis), termed as metabolic associated fatty liver disease (MAFLD) [1] or hepatic dyslipidemia. The process of MAFLD, accompanied by inflammation, cell death, and scarring (fibrosis), may lead to end-stage liver disease or hepatocellular carcinoma. MAFLD is not only intimately associated with diabetes and obesity [2] but also a risk factor for cardiovascular disease [3]. Although genetics, diet, nutrition, drugs, and other factors are all related to MAFLD, novel functional agents to effectively lower the fat accumulation in liver are worthy of further exploration.

Carotenoids are a group of isoprenoid compounds responsible for red to pink colors found in seeds, fruits, and vegetables. As antioxidant phytochemicals, they have a variety of biological functions, such as beneficial effects on eye health and improvements in cognitive function and cardiovascular health [4]. Carotenoids also show positive effects in inhibiting the development of diabetes and insulin resistance, obesity, non-alcoholic liver disease, and other diseases [5,6]. Torularhodin is a β-carotene -like compound from *Sporidiobolus pararoseus* [7], and it not only exhibits superior antioxidant properties with strong scavenging activity of peroxyl free radicals [8], but also has other beneficial effects, including improvement of neuroinflammation and cognitive impairment [9] and anti-cancer activity [10]. Furthermore, the LDL-lowering effect of Torularhodin in alcoholic liver injury in mice [11] and its protective function against oxidative-induced liver damage [12] indicate Torularhodin might also have an anti-hepatic dyslipidemia activity, which is worthy of exploration to increase its potential as a therapeutic agent to the treatment of MAFLD and related health complications. In the current study, a proteomic approach, accompanied by metabolite analysis, was applied to explore the effect of the Torularhodin on hepatic lipid metabolism in high fat diet-induced obese mice.

## 2. Materials and Methods

### 2.1. Materials

Torularhodin (purity > 95%) was isolated and purified from the extract of *Sporidiobolus pararoseus* (JD-2 CCTCC M 2010326) according to a previously published method [10]. Animal diets were purchased from TROPHIC Animal Feed High-tech Co., Ltd. Commercial kits for serum concentration of total cholesterol (TC), low-density lipoprotein cholesterol (LDL-c), high-density lipoprotein cholesterol (HDL-Cc), and triglyceride (TG) were purchased from Jiancheng Technology Co. (Nanjing, China). Hematoxylin and eosin (H&E) were obtained from Servicebio Technology Co. (Wuhan, China). Antibodies of PPARα, PPAR-γ, CYP7A1, CPT1A, SLC27A4, and GAPDH were obtained from ABclonal Technology Co. Ltd. (Wuhan, China). Western fluorescence detection reagent was obtained from Beyotime Biotechnology (Shanghai, China).

### 2.2. Animal and Treatment

The animal experiment was approved by the Institutional Animal Care and Use Committee of Jiangnan University (No.JN.No2020915c0501214[208]). Male C57BL/6J mice (11-week-old) were purchased from the SiLaiKe Laboratory Animal Co. (Shanghai, China). The mice were housed with free access to water and food ad libitum in a standard specific-pathogen-free (SPF) facility under a controlled environment (temperature of 23 ± 2 °C, humidity of 60 ± 5%, and 12:12 h dark/light cycle). After acclimatization for one week, the mice (BW: ~26.4 g) were randomly divided into three groups (*n* = 10/group): the control group, HFD group, and HFD-T group. The control group was fed on a normal chow diet (AIN-93M, 10% energy derived from fat, 3.6 total kcal g^−1^). The high fat diet (TP 23100, 45% high fat diet, 4.5 total kcal g^−1^) for the HFD group was from TROPHIC Animal Feed High-Tech Co. Ltd., China. The HFD-T group was fed on a HFD containing Torularhodin with a content corresponding to a dose of 40 mg/kg/day (45% high fat, 4.5 total kcal g^−1^). All animals were treated on different diets for 12 weeks. The doses of Torularhodin chosen in this current experiment were based on our previous studies [12].

The body weight and food intake of the mice were monitored weekly. At the end of treatment (12 weeks), after overnight fasting (water was allowed), all mice were sacrificed under anesthesia by inhalation of isoflurane. The collected blood was centrifuged at 2300× *g* for 15 min to obtain the serum. The harvested liver tissue was rinsed with PBS buffer, weighed, and immediately immersed in liquid nitrogen. All samples were stored at −80 °C until analysis.

### 2.3. Phenotype Parameter Analysis

The serum concentration of total cholesterol (TC), low-density lipoprotein cholesterol (LDL-C), high-density lipoprotein cholesterol (HDL-C), triglyceride (TG), TNF-α, IL-1β, IL-6, and lipopolysaccharide (LPS) were determined based on the corresponding instructions of assay kits purchased from Nanjing Jiancheng Bioengineering Institute (Nanjing, China). The liver parts from six mice were cut into pieces and fixed in 4% paraformaldehyde for 24 h. After the samples were dehydrated in ethanol, they were embedded in paraffin and cut into 5-μm thick sections. Finally, the sections were stained with hematoxylin and eosin (H&E), and the images were acquired by light microscopy (Olympus, Tokyo, Japan) (×200).

### 2.4. Proteomics Experiment

#### 2.4.1. Hepatic Protein Sample Preparation

The liver tissue was first ground into powder in liquid nitrogen, and to lessen the variation between mice, liver tissues from three mice were pulled together as one sample (total 3 samples) for the following analysis. The lysis solution (7M urea, 2M thiourea, 0.1%CHAPS, 1 tablet protease and phosphatase inhibitors (Thermo Fisher Scientific, Waltham, MA, USA)) was added into the sample (1 g/10 mL) to lysis the liver tissues for 30 min. After centrifugation (20,000× *g*, 30 min, 4 °C), a BCA protein quantification kit (Fisher Scientific, MA, USA) was used to measure the content of protein in the supernatant. Then, the SDS-PAGE was performed to observe the degree of lysis. For the preparation of the protein sample for Mass Spectroscopy, 100 μg protein solution was pipetted into a new EP tube containing 100 μL ammonium bicarbonate (100 mM) and DTT (10 mM). After incubation in a water bath (56 °C) for 1 h, iodoacetic acid (IAA, with a final concentration of 50 mM) was added in a dark room for 1 h. After the protein sample was cleaned by TEAB solution and centrifuged at 20,000× *g* for 30 min, 100 μL ammonium bicarbonate (100 mM) was added and mixed for proteolysis (1.0 μg trypsin per 100 μg of protein) at 37 °C for 12 h. After further cleaned with TEAB solution, the standard TMT labeling was applied to the peptides. Finally, the peptide solution was freeze-dried (alpha1-2LD; Christ, Osterode, Germany) for further analysis.

#### 2.4.2. Peptide Desalination

The desalting sep-Pak C18 column (Waters, MA, USA) was activated with a 200 μL solution composed of trifluoroacetic acid (TFA, 0.1%) and acetonitrile (60%). After the column was fully equilibrated using the equilibration solution (400–600 μL) composed of 0.1% TFA and 1% acetonitrile, the digested peptide solution was loaded onto the column, and then the column was washed 3 times with a solution of 0.1% TFA and 0.5% acetonitrile to remove the salts. Finally, 300 µL elution buffer (0.1%TFA, 60% acetonitrile) was applied to the column to elute peptides, and the eluted solution was freeze-dried to remove acetonitrile.

#### 2.4.3. HPLC-Mass Spectrometry

The TMT-labeled peptides were dissolved in solution A (98% ddH_2_O, 2% acetonitrile, and ammonia water to adjust the pH to 10.0), and then an Offline XBridge^®^ peptide BEH C18 high performance liquid chromatography column (130Å, 3.5 μm, 4.6 mm × 150 mm) (RP-HPLC) was used to separate the labeled peptides. The elution buffer B (98.0% acetonitrile, 2.0% deionized water (pH 10.0)), in combination with solution A in a gradient, was used to elute the peptides into multiple fractions (based on different gradient of elution buffer). The collected fractions were freeze-dried for the following Mass spectrometry.

The peptide fraction was dissolved in 10 μL of 0.1% formic acid solution. An on-line, low pH, reverse-phase C18 capillary chromatography (75 μm × 150 mm, 3 μm) was used to separate the peptides with a 5–38% gradient elution composed by phase A (99.9% H_2_O and 0.1% FA) and phase B (100% ACN, 0.1% FA). The total elution time was 120 min at a flow rate of 0.5 μL/min. Then, the peptide mixture was analyzed and detected using the Orbitrap Fusion™ Tribrid™ mass spectrometer (Thermo Scientific, Waltham, MA, USA). The Mass Spectrum acquisition mode was set to high sensitivity, and a high-speed signal-dependent scan was performed under the standard conditions.

#### 2.4.4. Mascot Qualitative-Quantitative Analysis

The resulting mass spectral data were retrieved by Mascot (version 2.8.0) (Matrix Science Inc., Boston, MA, USA from the database of SwissProt_mouse (Download time, March 2021). The search parameters are: trypsin up to 2 missed cleavage sites, the mass errors of the parent ion and fragment ion are 10 ppm and 0.05 Da, respectively. The fixed modification is Carbamimethy (C), and the variable modification includes Oxidation (M) and Acetylation at the N-terminal. The protein sequence data refers to the sequence translated from transcriptome sequencing results. Data quality control and quantification were processed by Scaffold Q+ (version 4.5.3) (Proteome Software, Inc. Portland, OR, USAThe false discovery rate (FDR) of peptides and proteins is less than 1.0%, and at least 2 specific peptides have been identified for each protein, with normalization based on the median data.

### 2.5. Hepatic Metabolite Analysis

The metabolite analysis was based on literature reports [13,14] in sample preparation and metabolite analysis. Briefly, an Ultra-high performance liquid phase tandem Thermo Q Exactive Orbitrap mass spectrometer, which was equipped with a UPLC HSS T3 column (2.1 mm × 100 mm, 1.8 μm) (Waters Corporation, Shanghai, China), and the control software (Xcalibur, version: 4.0.27, Thermo Scientific (Waltham, MA, USA)) was used as the liquid-mass system for primary and secondary mass spectrometry data acquisition. The mobile phase under a positive ion mode was composed of A: 0.1% formic acid aqueous solution and B: acetonitrile. For negative ion mode, mobile phase A was 5 mmol/L ammonium acetate aqueous solution and mobile phase B was acetonitrile. MS-DIAL [15] software was used to identify the metabolites based on the open-access database of MetaboLights (https://www.ebi.ac.uk/metabolights (accessed on 9 October 2021)).

### 2.6. Western Blotting Analysis

The liver tissue was homogenized in a RIPA lysis buffer containing inhibitors of phosphatase and protease (ThermoFisher Scientific, Shanghai, China), and the concentration of total protein in the supernatants after centrifugation was analyzed using a BCA kit (Shanghai, China). After protein samples were solubilized in an SDS sample buffer and heated for 10 min at 95 °C, an equal amount of protein was used in electrophoresis (NuPAGE 4–12% Bis-Tis GEL; Thermo Fisher Scientific). After the protein was electro-transferred onto polyvinylidene difluoride membranes (IBLOT2, Thermo Fisher Scientific), the membranes were first blocked with TBST containing 5% nonfat milk powder for 2 h at room temperature. Then, the blocked membrane was incubated (4 °C overnight) with a suitable dilution of primary antibodies solution of PPARα, CYP7A1, CPT1A, SLC27A4 and GAPDH followed by incubation with the secondary antibody solution (2 h, room temperature in the darkroom). After the blotted membrane was treated for 2 min in an ECL working solution, the blot was photographed in an imaging system. The luminous intensity representing the content of the protein was analyzed using ImageJ software (Bethesda softworks LLT, Bethesda, MD, USA), and the relative amount of protein was calculated and normalized based on housekeeping protein of GAPDH.

### 2.7. Statistical and Bioinformatics Analysis

All experimental results were presented as the mean ± standard deviation (SD). One-way analysis of variance (ANOVA) with a post hoc test was conducted using data processing software (SPSS version 22, IBM) ( New York, NY, USA). A Pearson correlation among all the phenotypic parameters was also conducted, and a *p*-value of <0.05 was considered statistically significant. The bioinformatics analysis, which includes PCA, heat map of the differentially expressed protein (DEPs) and metabolites, the COG/KOG annotation, KEGG (Kyoto Encyclopedia of Genes and Genomes) signaling pathway, protein–protein interaction (PPI), and Spearman’s correlation between phenotypic parameters and DEPs or metabolites, was performed based on standard procedures in R program.

## 3. Results

### 3.1. Anti-Obesity and Anti-Hepatic Dyslipidemia Effect of Torularhodin

To understand the impact of Torularhodin on lipid metabolism, the changes of body-weight and serum lipid profile, as the phenotype parameters, were examined. After mice (with similar bodyweight at the beginning) (26.4 g) were treated for 12 weeks, a statistically significant difference in the bodyweight of mice (*p* < 0.01) was observed among groups fed on control (32.05 g), HFD-T (39.89 g), and HFD (44.93 g) (Figure 1A) along the food intake (Figure 1B). Consistently, the serum lipid content was also significantly affected by the treatments (Figure 1C–F). The highest contents of serum triacylglycerol (TG, 1.6 mM), total cholesterol (TC, 7.5 mM), and low-density lipoprotein cholesterol (LDL-c, 1.05 mM) were observed in the HFD group, but they were substantially reduced by 24.5%, 25.3%, and 33.3%, respectively, in the HFD-T group. On the contrary, the content of the high-density lipoprotein cholesterol (HDL-c, 4.2 mM) in the HFD-T group almost approached the level of the control group, while the HFD group had the lowest content (Figure 1F). Further correlation analysis showed that the body weight is significantly correlated with TG, TC, LDL, and HDL with correlation coefficients of 0.63, 0.71, 0.63, and −0.56, respectively, indicating the lipid dyslipidemia is intimately associated with HFD-induced obesity, and the addition of Torularhodin significantly alleviated the adverse effects of HFD on lipid homeostasis and elevated the content of the health-promoting HDL-c. The health benefits of Torularhodin were also evidenced by increased insulin sensitivity with lower level of fasting serum insulin (Figure 1G) and fasting blood glucose (Figure 1H) in HFD-T group. No significant differences in energy intake between the HFD and HFD-T groups (Figure 1B) indicate that the dosage of Torularhodin used in the current study is not high enough to produce adverse effects.

Liver and adipose tissue H&E staining (Figure 2A) was performed to examine the lipid accumulation. The representative photomicrographs showed that the lipid vacuoles of the HFD-fed mice were dramatically increased compared with the HFD-T group. This result not only signifies the pivotal role of the liver in lipid metabolism, but also confirms the preventive function of Torularhodin against the lipid accumulation in hepatocytes as well as in the adipose tissue (Figure 2B). The reduced body weight through a reduction of fat accumulation in adipose tissue may be directly associated with the health benefits of Torularhodin.

### 3.2. Proteomics and Metabolite Profiling

#### 3.2.1. Lipid Metabolism-Related Hepatic DEPs

To elucidate the mechanism of Torularhodin in alleviating HFD-induced hyperlipidemia, proteomics was used to identify differentially expressed proteins (DEPs) in the liver tissue was performed. A total of 3012 differentially expressed proteins were identified and quantified by labeling experiments. When the DEPs were screened according to the fold changes (fold change ≤ 0.67 or ≥1.5, student’s *t*-test *p* < 0.05 ), HFD-T (compared to the control) induced 193 DEPs in which 133 were up-regulated and 60 were down-regulated (Figure 3A). When compared to HFD, HFD-T showed a total of 512 DEPs in which 223 were up-regulated and 289 were down-regulated (Figure 3B), indicating significant changes of gene expression in HFD-T group. The principal component analysis (PCA) also clearly distinguished the three groups (Figure 3C).

When the expression levels of lipid metabolism-related DEPs were compared among all the groups, the protein profile of HFD and HFD-T showed almost opposite trend to each other whereas HFD-T is more similar to that of the control (Figure 3D). A high expression of proteins favoring lipid degradation accompanied by a down-regulation of proteins related to lipid accumulation may ultimately improve lipid homeostasis. Certainly, the up-regulated proteins in the HFD-T group (compared to HFD) mainly include β-oxidation-related enzymes (CPT1A, ECI2, ACAA1A, ACAA1B, NDUFS8, GK) and fatty-acid binding apolipoproteins (APOA-I, APOA-II) related to beneficial HDL-c. The increased content of CYP7A1, converting cholesterol to bile acids for the excretion, suggests that cholesterol catabolism is another important mechanism to improve the dyslipidemia. The up-regulated proteins of PCK1 indicate that the glucose metabolism is also involved in dyslipidemia. Processes related to protein and amino acid metabolism (TMEM30a, EMC1, RPS10) with different functions are also significantly increased. Thus, the up-regulated DEPs (Appendix A) promote FFA β-oxidation, bile acid synthesis, phospholipid balance, methyl cycling, and glucose homeostasis. The protein and enzymes related to amino acid metabolism and proteins processing (e.g., ensuring the distribution pattern of phospholipids in membrane) may aid the lipid homoeostasis through different ways.

The proteins downregulated by Torularhodin mainly include SLC27A5, FABP1, GLOD4, VNN1, ME1, PLIN2, FABP2, PPM1K, SLC27A4, PLIN5, PLIN4, and ASL (Appendix A). Functionally, SLC27A5, ME1, FABP2, FABP1, and SLC27A4 are involved in fatty acid uptake and transport. PLIN2, PLIN4, and PLIN5 are perilipin family members that coat intracellular lipid storage droplets. A recent report also showed that a high level of PLIN2 protects lipid droplets against autophagy, whereas PLIN2 deficiency (similar to a reduced level of PLIN2) enhances autophagy (lipophagy) and depletes hepatic TG [16], which might be consistent with the up-regulated VAMP8 that is a SNARE protein in autophagy. Amino acid metabolism-related proteins of PPM1K (branched-chain amino acid catabolism) and ASL (L-arginine biosynthesis and the urea cycle) represent another category of down-regulated proteins. Thus, most of the proteins down-regulated by Torularhodin are involved in lipid uptake and transport, fatty acid biosynthesis, storage, and unbalanced amino acid metabolism, and their down-regulation is favorable to lipid homeostasis.

#### 3.2.2. Other Metabolic Processes-Related Hepatic DEPs

Given the complexity of obesity, processes related to ketone body (3-hydroxybutyric acid), betaine metabolism, and other transport proteins were further analyzed. The lower levels of PPARα target enzymes of ACAT1 (Acetyl-CoA Acetyltransferase), BDH (D-beta-hydroxybutyrate dehydrogenase), and HMGCS2 (3-hydroxy-3-methylglutaryl-CoA synthase) that catalyzes the first reaction of ketogenesis demonstrated that the addition of Torularhodin reduced the ketogenic property of HFD, which is consistent with the increased expression of PCK1 that is involved in glucose metabolism. The increased protein level of BBOX1 (Gamma-butyrobetaine dioxygenase) that catalyzes the formation of L-carnitine (critical molecules in β–oxidation by transporting long-chain fatty acids into the mitochondria) from gamma-butyrobetaine is consistent with the lipid degradation. The up-regulated BHMT (Betaine--homocysteine S-methyltransferase 1) [17], which catalyzes the conversion of betaine and homocysteine into dimethylglycine and methionine, also exhibits the health function of Torularhodin with decreased level of homocysteine, which is a risk factor of cardiovascular disease. Additionally, many ATP-binding cassette (ABC) transporters [18] are also upregulated, such as ABCB4 which plays a role in the phospholipid (PC, PE, SM) arrangement in the membrane and the proper phospholipid bile formation for cholesterol efflux in the presence of bile salts. ABCB7, a mitochondrial iron transporter controlling intracellular iron homeostasis, can inhibit both apoptotic and non-apoptotic cell death by reducing mitochondrial ROS and activating nuclear factor-kappa B signaling [19]. Highly upregulated ABCBs also include ABCB8, ABCB10, and ABCB11. ABCB8 is involved in ATP-dependent potassium currents in the mitochondria. ABCB11 is a bile salt export pump to remove hepatic bile salts. Conclusively, the addition of Torularhodin caused numerous physiological changes through regulating protein expression.

Clusters of Orthologous Genes (COG) category analysis was conducted to further understand the functions of DEPs. Clearly, broad changes, including almost all categories in the COG system, were observed. For the COG in Control vs. HFD (Figure 4A), the highly-enriched COG categories include **C**: Energy production and conversion, **I**: Lipid transport and metabolism, **O**: Posttranslational modification, protein turnover, chaperones, **T**: Signal transduction mechanisms, and **U**: Intracellular trafficking, secretion, and vesicular transport. For HFD-T vs. HFD, the enriched COGs mainly include **A**: RNA processing and modification, **J**: Translation, ribosomal structure and biogenesis, category **O**, **T**, and **U**. It can be seen that both Torularhodin treatment and HFD induced significant changes at the COG level, and protein synthesis and signal transduction are likely the dominant COG categories. On the other hand, since both COG analyses are relative to HFD, a comparison between them could indirectly illustrate the function of Torularhodin. Indeed, a correlation coefficient of 0.78 (*p* < 0.01) was obtained between both groups (Control vs. HFD, HFD-T vs. HFD) when all the COG category and matched frequency were analyzed, indicating the similarity between HFD-T and control group. Torularhodin may recover a larger portion of the physiological processes adversely affected by HFD through regulating gene expression.

#### 3.2.3. Hepatic Metabolite Profiling

In order to enhance the understanding of the physiological processes affected by HFD and Torularhodin, the metabolites were also analyzed using the UPLC system. A comparative analysis of the changes in liver metabolites based on the screening condition (*t*-test, *p* < 0.05, fold change ≥1.5 or ≤0.67) showed distinct metabolite profiles among the groups (Figure 5A,B), and three groups were separated from PCA analysis (Figure 5C). Specifically (Figure 5D), compared with the control group, the HFD group produces high content of fatty acid metabolites (such as palmitoleic acid, eicosenoic acid, eicosapentaenoic acid), amino acids and their metabolites (betaine, L−lysine, L−tryptophan and its metabolite of indoxylsulfuric acid, L−methionine, L−Phenylalanine, L−Proline), phospholipids (PE19:0−18:2, LPC18:3, PS18:0−22:6, PE17:0/0:0, LysoPE22:6, PC22:6, PC20:4), and purine metabolites (ADP-ribose, hypoxanthine, xanthine, inosine), which is consistent with the literature report [20]. In the meantime, lipid degradation-related metabolites (butyryl−CoA, acetyl−CoA, L−carnitine) and bile acids (glycocholic acid, taurine, tauroursodeoxycholic acid, taurocholic acid, cholic acid methyl ester) were significantly reduced. For HFD-T group, when compared to HFD group, the opposite changes of most metabolites were observed. Thus, the addition of Torularhodin increased FFA oxidation, bile acid production, and cell membrane integrity, which is consistent with the outcomes from proteomics analysis.

### 3.3. The Correlation between Phenotypic Parameters and Hepatic Proteins/Metabolites

The relationship between the phenotypic parameters and omics data is essential to reveal the underlying mechanism of Torularhodin’s beneficial functions. As shown in Figure 6, an opposite trend is observed between HDL-c and other phenotypic parameters of body-weight, TG, TC (Figure 6 left panel), indicating the elevation of HDL-c is intimately associated with the antihyperlipidemic effect of Torularhodin. Indeed, HDL-c is beneficial to health with its cholesterol-lowering function and inverse correlation with atherosclerotic cardiovascular disease [21]. Additionally, HDL also exhibits antioxidant and anti-inflammatory effects [22]. Thus, the elevation of HDL-c to transport cholesterol to the liver for excretion might be one important pathway for Torularhodin to exert its antihyperlipidemic functions. Further analysis of the DEPs showed that HDL-c is positively associated with proteins GULO, CYP7A1, APOA1, DDC, RPS10, SRRT, UGT2A3, SEC61B, LSR, and ASGR1. With the function of bile acid biotransformation from cholesterol, CYP7A1 is also associated with HDL metabolism [23]. Apo-AI is the structural protein of HDL [24]. Ribosome protein RPS10 and gene expression-related SRRT and SEC61B (protein translocation) are also associated with HDL-c, indicating that complex protein synthesis is involved in HDL-c formation. ASGR1 is an essential protein to serum glycoprotein homeostasis through endocytosis and lysosomal degradation, and a literature report has shown that it is associated with catabolism of low-density lipoprotein [25]. LSR (lipolysis stimulated lipoprotein receptor) is involved in the clearance of triglyceride-rich lipoprotein, and knockdown of LSR promoted hypertriglyceridemia and increased serum levels of apolipoprotein (Apo)B and E [26], which are associated with LDL and chylomicrons. The analysis of the functions of these positively associated proteins showed that the protein processes and the metabolism of lipoprotein likely represent the underlying mechanism. As for the negatively correlated proteins (Fabp2, Slc27a4, Me1), they are mainly involved in lipid uptake and lipid biosynthesis, which indicates hyperlipidemia might inhibit the formation of HDL-c.

In order to complement the proteomics, the metabolite profile was analyzed. Similar as the proteomics, distinct metabolite profiles were observed between HDLc and other parameters (Figure 6 right panel). The positively correlated metabolites with HDLc at a significant level include acetyl-CoA, butyl-CoA, betaine, L-carnitine, tauroursodeoxycholic acid, and propionyl−CoA, while the negatively correlated metabolites include 9−HODE, xanthine, and lysophosphatidylcholine 20:4. Thus, fatty acid synthesis, purine metabolites, and reduced membrane integrity all contribute to dyslipidemia with a reduced level of HDL-c. Actually, the ratio between uric acid (end product of purine metabolism) and HDL is a biomarker of metabolic associated fatty liver disease (MAFLD) [27], indicating that the purine metabolite might disrupt the formation of HDL.

### 3.4. Anti-Hepatic Inflammation by Torularhodin

It is well known that obesity is intimately associated with a chronic low-grade inflammation [28]. Our current study also demonstrated high levels of inflammatory cytokines, including TNF-α, IL-6, and IL-1β, in HFD group while the addition of Torularhodin into the HFD significantly reduced the levels of these cytokines (Figure 7A–C). Pearson correlation among all the phenotypic parameters (Figure 7E) showed that only HDL-c exhibits a significantly negative correlations with all other parameters. However, parameters of dyslipidemia (TC, TG, LDL-C, fasting Insulin) and inflammation (TNF-α, IL-6, IL-1β) are highly correlated, indicating a significant association between dyslipidemia and inflammation, which is consistent with the correlation heatmap showing a similar pattern between cytokines and lipid metabolism-related parameters (Figure 6). It is also noted that a high level of LPS, an endotoxin derived from the outer membrane of Gram-negative bacteria, was detected in HFD group (Figure 7D) and a stronger correlation was found between LPS and other parameters (Figure 7E). The presence of LPS suggests that imbalanced microbiota and barrier function impairment are also related to obesity, and reduced obesity-induced inflammation and circulated LPS by Torularhodin might be another aspect of improved health.

Further proteomics analysis identified many proteins that are positively correlated to the cytokines, particularly in the HFD group (Figure 8A). However, an opposite pattern of proteins in the heatmap was observed in HFD-T group. Thus, the inflammation is also closely related to the proteins involved in the immune system (Appendix A). Specifically, anti-inflammatory proteins GSDMD, FAS, BAX, ICAM1, OCLN, GSTP1, FAF1, LRP1, APEX1, ROCK1, MANF, STAT3, and INSR were significantly upregulated in HFD-T group while the pro-inflammatory proteins OPTN, PTK2B, FADD, MIF, CASP3, YAP1, DNM1L, and NAMPT were all downregulated. Clearly, multiple immune processes are involved in the obesity-induced inflammation including cell apoptosis, NF-κB signaling pathway, cytokine synthesis, and receptor-mediated actions. A schematic presentation of the action of these proteins is depicted (Figure 8B) based on the function of identified proteins.

### 3.5. Hepatic Protein-Protein Interaction

To further the understanding of the DEPs under the treatment of Torularhodin. The protein–protein interaction was analyzed (Figure 9). The PPI analysis showed that there are two major networks of lipid metabolism and inflammation, which are linked through Lrp1 (LDL Receptor Related Protein 1), Lsr (Lipolysis Stimulated Lipoprotein Receptor), and Pck1 (Phosphoenolpyruvate carboxykinase 1), indicating an intimate relationship between lipid metabolism and inflammation as well as the glucose metabolism. The lipid-centered PPI showed that lipid metabolism, fatty acid degradation, β-oxidation, bile synthesis and excretion, apolipoprotein metabolism, amino acid metabolism, glucose metabolism, as well as their binding activity and transport, were interconnected to each other. The proteins APOA1, FABP1, CPT1A are the main nodes in the network with the highest degree of connectivity, which can be considered as the cornerstone of the network. The protein HMGCS2 representing ketogenesis is another important node connected with lipid metabolism. Regarding the inflammation-centered PPI, biological processes of mitochondria fragmentation in apoptosis, T-cell homoeostasis, B-cell apoptotic process, chemokine production, ATP metabolism, lymphocyte activation in immune response, unsaturated fatty acid metabolism, and mitochondria autophagy are all interconnected in the inflammation process. The protein CASP3 and STAT3 are the nodes with the highest degree of connectivity, which indicates the cellular apoptosis and inhibition of inflammation are the main processes affected by Torularhodin to achieve an immune balance. The cell surface receptors acting as receivers of intrinsic and extrinsic signals are also important nodes in the inflammation-centered PPI network. Signaling pathway analysis from the PPI reveals that the PPARα signaling pathway is the main pathway in lipid-centered processes while the FAS-pathway, Toll-like receptor pathway, and one carbon cycling pathway are related to the immune response. Thus, these PPI data not only provides an overall picture of the proteins involved in lipid homoeostasis and improved inflammation, but also demonstrate the cooperation patterns and orders among all the proteins in responses to Torularhodin treatment.

### 3.6. KEGG Pathway Enrichment Analysis

KEGG pathway enrichment analysis based on DEPs revealed multiple signaling pathways associated with Torularhodin (Figure 10A,B). Among the top 10 pathways obtained from protein enrichment, the metabolic pathways related to amino acids, phospholipids (sphingolipids, glycerolphospholipids), purine, fatty acid, pyrimidine, vitamin B6, glutathione, one-carbon cycle, nucleotide sugar, and taurine are the most highly and significantly enriched (Appendix A). Further analysis revealed that PPAR signaling is also significantly enriched, and due to its important function in regulating energy metabolism, cell differentiation, and inflammatory response [29], it might be the central pathway to mediate the beneficial effect of Torularhodin.

PPARs are transcription factors regulating gene expression following ligand activation [30]. Many PPARα target genes are involved in fatty acid metabolism, particularly hepatic fatty acid and plasma lipoprotein metabolism, which are intimated associated with obesity and dyslipidemia [31]. By contrast, PPARγ promotes fatty acid uptake, triglyceride formation, and storage in lipid droplets, as well as regulation of the proliferation and differentiation of cells, including adipose cells [32]. However, the analysis of proteomic data (PLIN4, ME1, APOA2, FABP1, PLIN2, FABP2, CPT1A, FABP5, SLC27A5, SORBS1, PLIN5, ACAA1B, SLC27A4, ACAA1A, CYP7A1, and PCK1 are all PPARα target genes) and metabolites showed that PPARα is the main signaling pathway for Torularhodin to exert its antihyperlipidemic function. The PPARα not only mediates the lipid degradation and bile acid synthesis, but also upregulates the expression of ABCB11 for bile excretion. The activation of PPARα also increased plasma HDL-c, facilitating the efflux of cholesterol via the expression of apolipoprotein A-I and apolipoprotein A-II [33]. Since PPARs are ligand-inducible transcription factors, Torularhodin and/or its metabolites might act as the ligand of PPARα to promote the expression of genes related to lipid metabolism and energy balance.

To validate the proteomics results and the PPARα signaling pathway, Western blot is necessary to verify the expression of the key proteins involved in the signaling pathway. Since β-oxidation, bile synthesis, HDL-c, and lipid transport are the major processes mediating the function of Torularhodin, the key proteins CYP7A1 and CPT1A, which are associated with PPARα-regulated fatty acid degradation, and SLC27A4, involved in PPARα-regulated fatty acid uptake and processing [34], were used in the Western blot. The results (Figure 11) showed, compared to the HFD group, a higher expression of PPARα, CYP7A1, and CPT1a in the HFD-T group. On the contrary, a decreased level of and SLC27A4 was observed in the HFD-T group. Thus, a reduction of lipid uptake from the small intestine and lower level of lipid droplet combined with increased fatty acid degradation would improve the health status of mice fed on HFD-T diet, and the PPARα signaling pathway activated by Torularhodin is likely the predominant mechanism.

## 4. Discussion

The proteomics analysis revealed that the PPARα-signaling pathway is the underlying mechanism of Torularhodin’s anti-hepatic dyslipidemia effect, which is consistent with the literature report on the function of PPARα on dyslipidemia [35] and further supported by a recent report on the occurrence of MAFLD after the hepatocyte-specific Pparα was deleted [36]. The activation of PPARα-target genes increased fatty acid β-oxidation, reduced fatty acid absorption, enhanced the conversion of cholesterol to bile acids for secretion, and promoted the formation of HDL-c, which are all beneficial to lipid homoeostasis and body weight reduction [37]. The reduced body weight may also contribute to the anti-hepatic dyslipidemia effect of Torularhodin.

PPARα is a crucial factor in lipid homeostasis as evidenced by its numerous target genes with functions of either lipid degradation and/or lipogenesis [34]. In our current study, PPARα activation leads to the up-regulation of lipid degradation-related enzymes and down-regulated proteins related to lipid uptake and synthesis. Different metabolites of Torularhodin, which may act as the ligands for PPARα activation, may have a different structure and affinity to bind PPARα. The opposite correlations between HDLc and other phenotypic parameters (TG, TC and LDL, TNF-α, IL-6, IL-1β, and LPS) indicate the importance of HDLc to lipid homeostasis. According to the reverse remnant-cholesterol transport (RRT) hypothesis [38], the major function of HDL is to transport cholesterol from remnant lipoproteins, which is produced during the lipolysis of triglyceride-rich lipoproteins (TGRL), to the liver for energy production or excretion through bile. Specifically, many positively correlated proteins (Figure 6), particularly PPARα targeted APOA1, which constitutes the skeleton of HDL, not only have a pivotal role in the HDL assembly process, but also act as the acceptor of cholesterol from cells [39]. The up-regulation of APOA1 may directly contribute to the increased content of HDLc in HFD-T group. Thus, the up-regulation of the HDL formation-related apolipoprotein also play an important role in Torularhodin’s health benefits.

An elevation of pro-inflammatory cytokines TNF-α, IL-6, IL-1β signifies the hepatic inflammation in HFD-induced obese mice. However, a significant reduction of these inflammatory factors was achieved in the Torularhodin-containing HFD-T group (Figure 7). In the meantime, a number of inflammation-related proteins or enzymes were identified. PPARα also plays an important role in the anti-inflammation property of Torularhodin through its target protein expression, such as inflammation-attenuating proteins ICAM-1 and STAT3. ICAM-1 is an adhesion receptor regulating leukocyte recruitment to sites of inflammation [40], and STAT3 is a transcription factor with anti-inflammatory function. Indeed, the literature has identified a number of PPARα target genes [34] involved in inflammation, such as Apolipoprotein A1 in anti-inflammatory HDL-c. Thus, all these data further support that PPARα is the master regulator of Torularhodin’s function. Additionally, PPARα signaling pathway also has an innate immune function in regulating intestinal inflammation, mucosal immunity, and commensal homeostasis through the targeted expression of immune function proteins, such as IL-22 and the antimicrobial peptides RegIIIβ [41]. The innate immune function of PPARα, once activated, will improve the intestinal barrier function and reduce the diffusion of endotoxin into the host. Thus, reduced inflammation is another important component for Torularhodin’s health function, and PPARα signaling pathway is also essential to its anti-hepatic inflammation properties.

## 5. Conclusions

In this study, we demonstrated that Torularhodin has an anti-obesity and anti-hepatic dyslipidemia effect through regulating the expression of the PPARα signaling pathway-related proteins to increase fat degradation and cholesterol execration accompanying reduced lipid absorption. The increased expression of apolipoprotein, particularly HDL-c-related ApoA-1 and ApoA-II, is also beneficial to improved dyslipidemia with its cholesterol-lowering function. Many proteins related to the anti-inflammatory function of Torularhodin are also PPARα target genes. Thus, PPARα signaling pathway is likely the fundamental mechanism of Torularhodin’s function of anti-hepatic dyslipidemia, anti-hepatic inflammation, and other health benefits. However, the interaction between key proteins and Torularhodin (or its metabolites) is still required to gain direct evidence of Torularhodin-mediated activation of the PPARα signaling pathway.

## Figures and Tables

**Figure 1 molecules-27-06398-f001:**
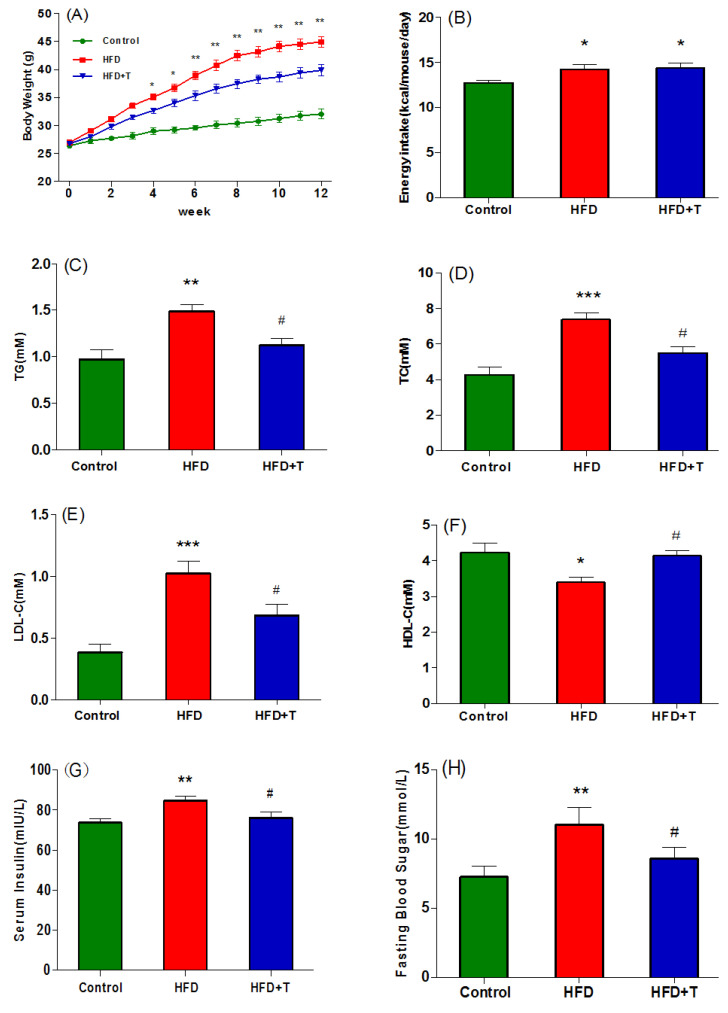
Phenotypical properties of mice in control group, HFD group and HFD-T group including (**A**) bodyweight, (**B**) energy intake, (**C**) serum TG, (**D**) serum TC, (**E**) serum LDL-C, (**F**) serum HDL-c, (**G**) fasting insulin, (**H**) fasting blood glucose TG. * *p* < 0.05, ** *p* < 0.01, *** *p* < 0.001 vs. group Control; # *p* < 0.05 vs. group HFD.

**Figure 2 molecules-27-06398-f002:**
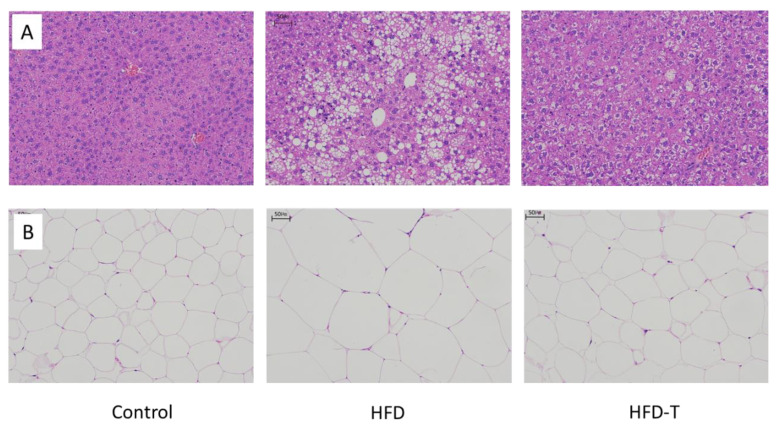
The morphology of liver and adipose tissue after treatment (number of staining = 6/group). Liver sections staining with H&E (scale bar 50 μm) (**A**), and white adipose tissue sections staining with H&E (scale bar 50 μm) (**B**). Control: normal chow diet, HFD: high fat diet, HFD-T: HFD containing.

**Figure 3 molecules-27-06398-f003:**
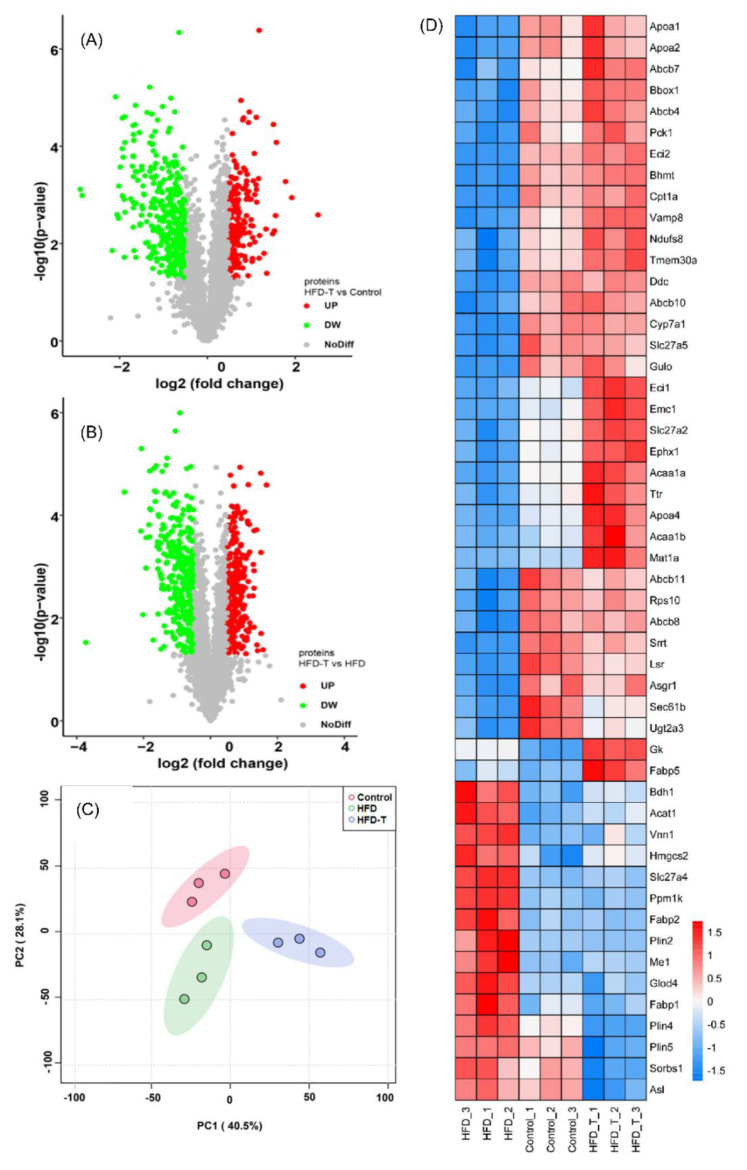
The volcano plot of DEPs (**A**,**B**) and the PCA analysis (**C**) under different treatment or diet as well as the heatmap of DEPs (**D**).

**Figure 4 molecules-27-06398-f004:**
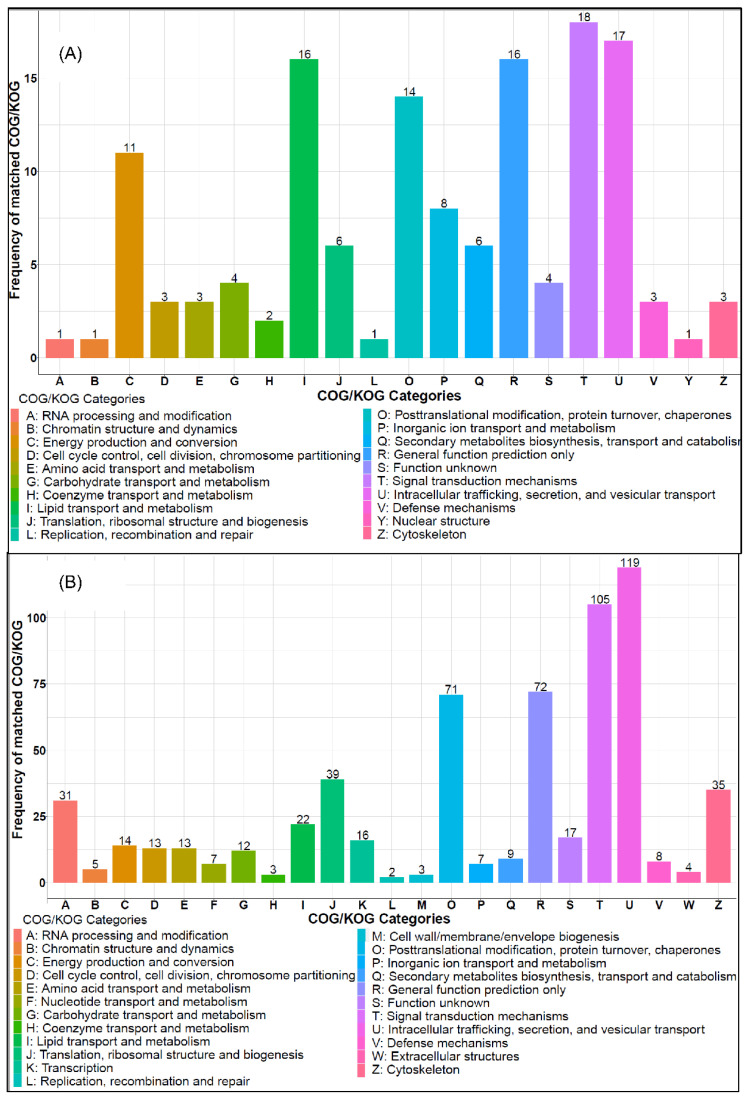
The COG/KOG analysis of DEPs in control (**A**) and HFD-T (**B**) group as compared to HFD group.

**Figure 5 molecules-27-06398-f005:**
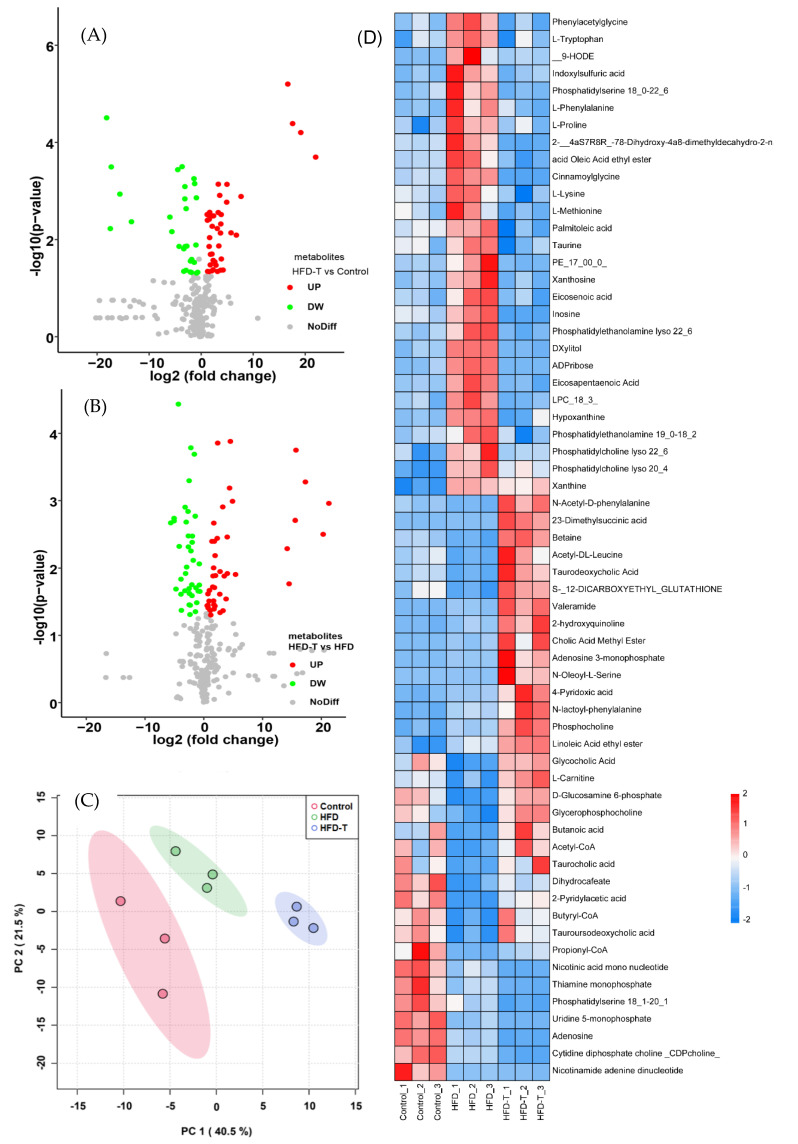
The volcano plot of metabolites (**A**,**B**) and the PCA analysis (**C**) under different treatment or diet as well as the heatmap of metabolites (**D**).

**Figure 6 molecules-27-06398-f006:**
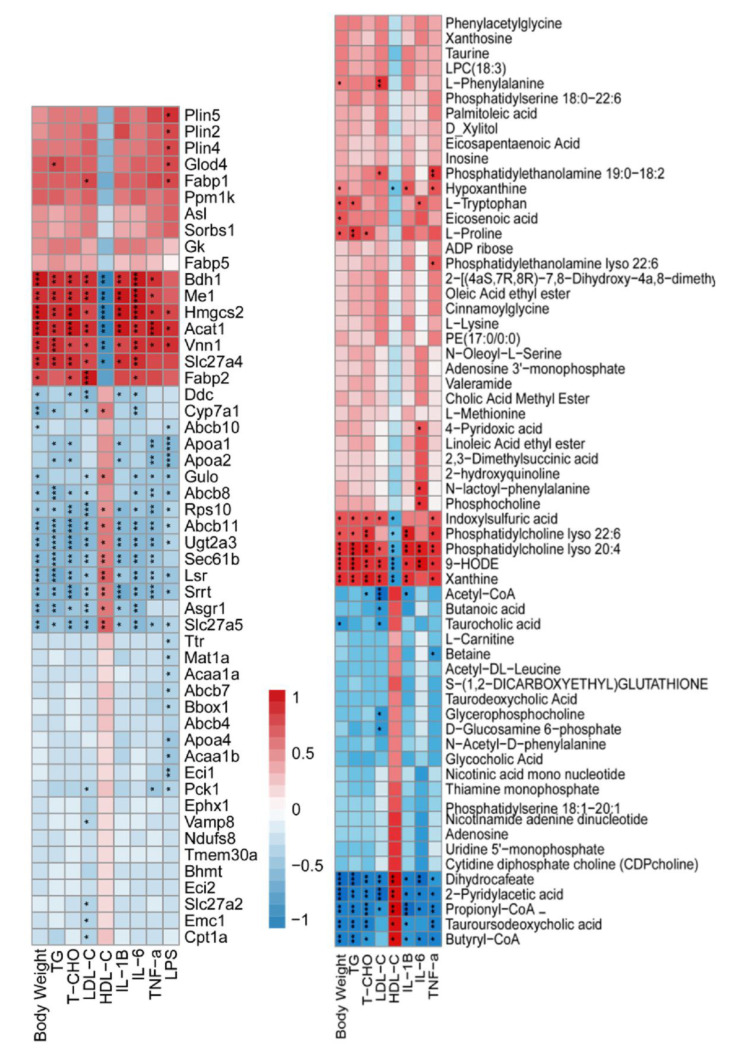
The correlation heatmap of the phenotypic properties of body weight, TG, TC, LDL-c, HDL-c, IL-6, IL-1β, and TNF-α with DEPs (**left panel**) and metabolites (**right panel**). The HDLc with unique association with DEPs and metabolites was demonstrated, and the significance of correlation as * *p* < 0.05, ** *p* < 0.01, and *** *p* < 0.001.

**Figure 7 molecules-27-06398-f007:**
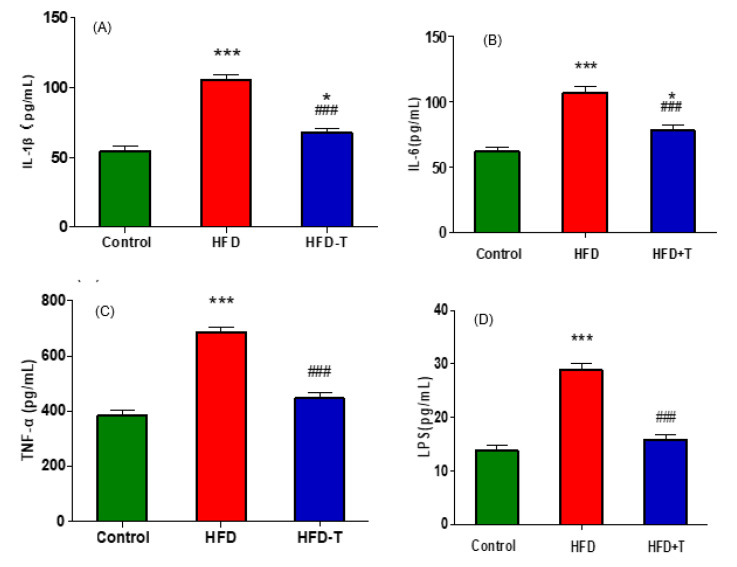
The anti-inflammation property of Torularhodin. Inflammatory cytokines IL-1β (**A**), IL-6 (**B**), TNF-α (**C**), and LPS (**D**). HFD-T compared to control: * *p* < 0.05, *** *p* < 0.001; HFD-T compared to HFD: ### *p* < 0.001. The Pearson correlation matrix of phonotypic parameters (**E**): **p* < 0.05, ** *p* < 0.01, *** *p* < 0.001.

**Figure 8 molecules-27-06398-f008:**
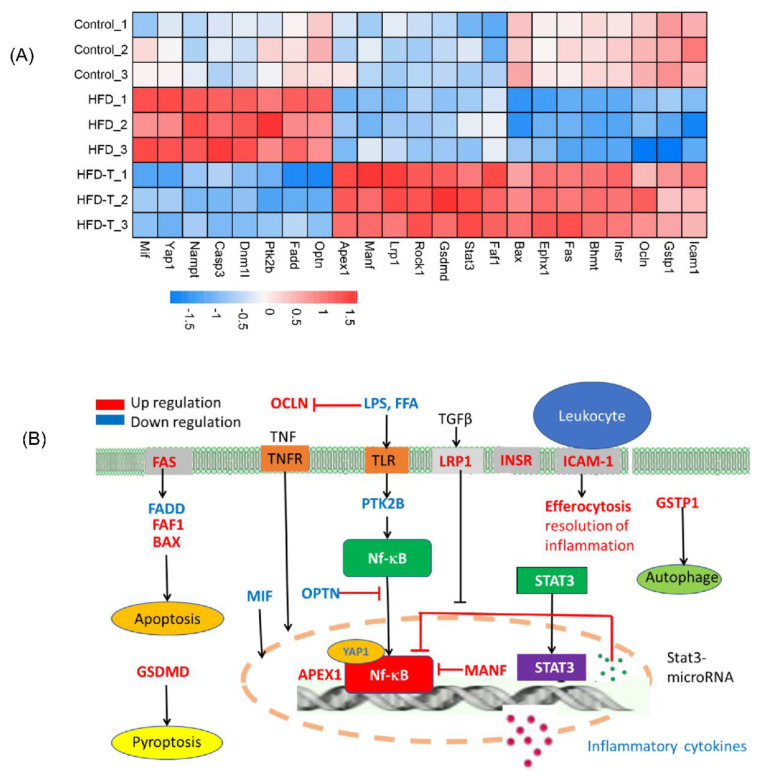
The anti-inflammation property of Torularhodin. The heatmap of inflammation-related proteins (**A**) and schematic illustration of the possible distribution of these proteins in the cells (**B**). Nf-κB and STAT3 are transcription factors.

**Figure 9 molecules-27-06398-f009:**
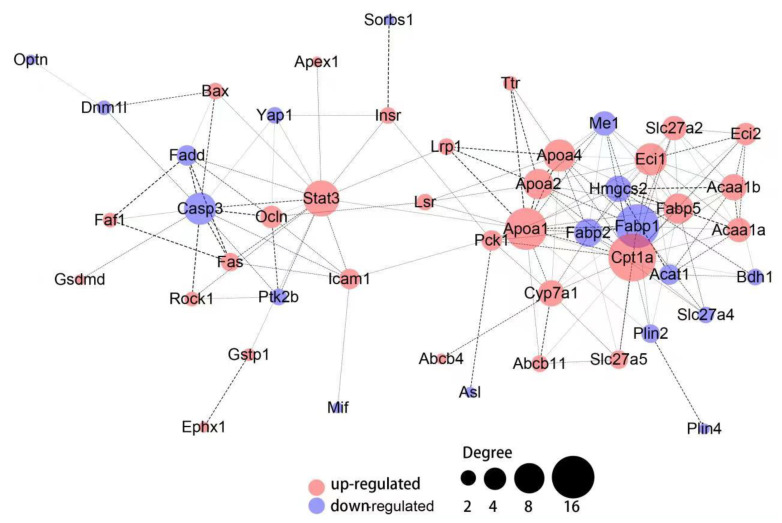
PPI analysis of the enriched proteins involved in lipid metabolism and inflammation under the treatment of Torularhodin. The width of the edge between the nodes represents the combined score from experimental, database annotated and data mining, the node degree represent the number of connected nodes.

**Figure 10 molecules-27-06398-f010:**
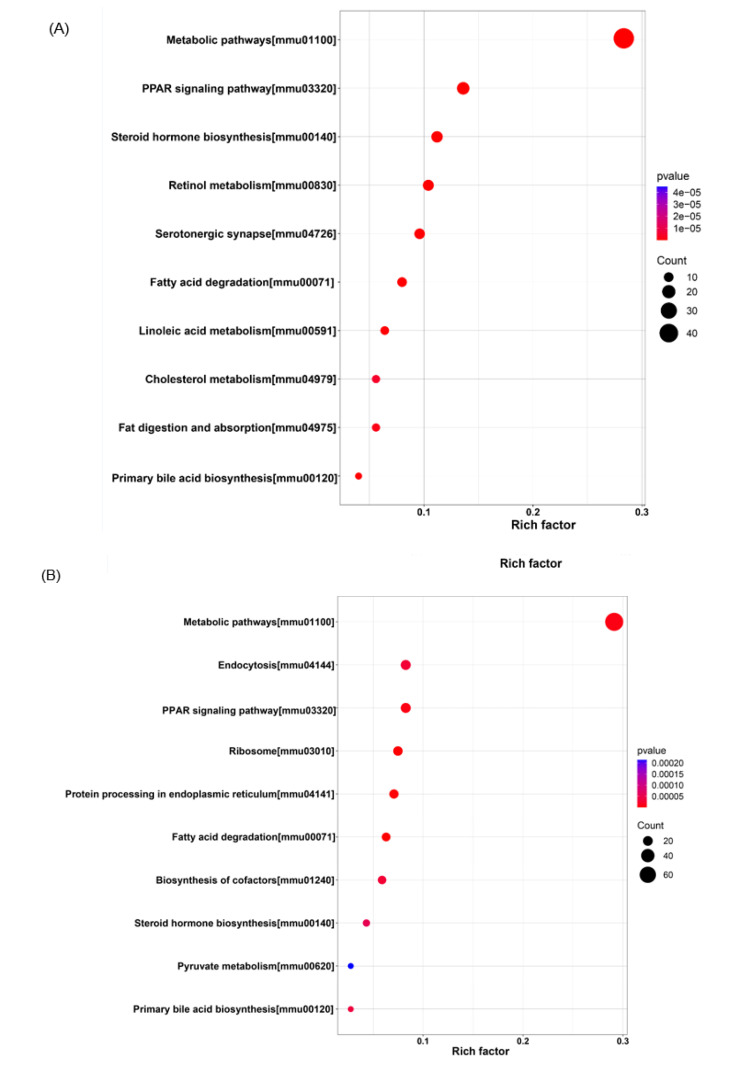
KEGG analysis of the signaling transduction pathways in Control vs. HFD (**A**) and HFD-T vs. HFD (**B**).

**Figure 11 molecules-27-06398-f011:**
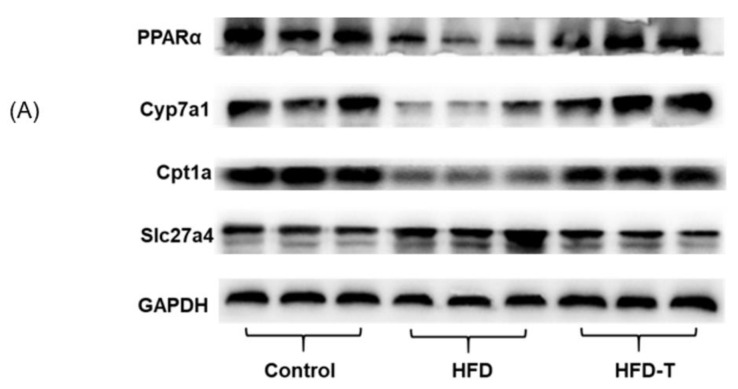
The expression of representative proteins affected by Torularhodin (vs. HFD group). (**A**): Representative photographs of western blots; (**B**): Relative expression levels of proteins from western blot (WB) and proteomics data (Proteomics). The measured data are expressed as mean ± SD (*n* = 3). *: *p* < 0.05, **: *p* < 0.01, ***: *p* < 0.001 (compared to HFD within the group).

## Data Availability

Not applicable.

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
