# Peer review of "Torularhodin Alleviates Hepatic Dyslipidemia and Inflammations in High-Fat Diet-Induced Obese Mice via PPARα Signaling Pathway"

_molecules, 2022, doi:10.3390/molecules27196398_

Round 1

Reviewer 1 Report

The manuscript used a proteomic and metabolite analysis to investigate the effect of torularhodin on lipid metabolism in high fat diet-induced obese mice. Overall, they performed a substantial amount of experiments and analyses and provided interesting data about the beneficial effects of Torularhodin on hepatic proteins and metabolites in the big picture. However, they are not well-organized in the manuscript and some interpretations of the results may be controversial. Overall, it is hard to understand the focus and conclusion of the manuscript. Therefore, major revisions are necessary to be accepted in this journal.

Major

1)      Based on the data from the manuscript, it is likely that the mechanisms underlying Torularhodin-mediated reduction in fatty liver and inflammation are dependent on body weight reduction. However, it is not represented in the manuscript, especially in the title and abstract.

2)      Similar to the first comment, it is hard to know whether the beneficial effects of torularhodin on the liver are a direct effect on hepatocytes or are dependent on body weight reduction. Discussion about this should be provided.

3)      Why do the authors use just one concentration of torularhodin? It would be better to give a rationale for it.

4)      The HFD-T group was fed on an HFD containing Torularhodin with 87 content corresponding to a dose of 40 mg/kg/day (45% high fat, 4.5 total kcal g1). How the dose of 40 mg/kg/day can be achieved if the mice were fed HFD mixed with Torularhodin instead of oral gavage feeding?

5)      How was the Correlation between Phenotypic Parameters and Hepatic Proteins /Metabolites calculated? What are the criteria for values? Detailed information about the methods and data interpretation should be included in the manuscript.

6)      To show triglyceride accumulation in the liver, H&E staining may not be enough. Oil red O staining or biochemical analysis of TG needs to be provided in figure 2.

Minors

1)      Line 206, (44.93 g)(Fig 1.A). – Add space between

2)      Line 207 (Fig 1 B, C, D&E) – In some places (line 221), multiple figures were mentioned using “and” instead of “&”. Be consistent throughout the whole manuscript.

3)      In figure 1, the detailed numbers of figure 1 (A, B, C, D) were not equally aligned.

4)      Figure 1g and figure 1h are derived from the same data. It is better to express the food intake as calories because the total calorie taken should be higher in the HFD group than in the control group. Better to delete figure 1g.

5)      More detailed information is necessary for figure 2 including N numbers for the staining

6)      In 249, what does PCA stands for?

7)      In line 268, does the proteomics performed using liver tissue? More detailed information is necessary for the figure legend

8)      Many typos and traces for grammatical editing are found in the manuscript (e.g. line 57, line 360, etc).

Reviewer 2 Report

Review for the manuscript “Torularhodin alleviates Hepatic Dyslipidemia and Inflammations in High-Fat Diet-induced Obese Mice via PPARα Signaling Pathway”

            Dear authors, it was a pleasure to review this article.

 I have some suggestions and corrections to be performed along with the text.

ABSTRACT

            In line 22 it is possible to see “…synthesis whereas the other phenotypic parameters (TC, TG, LDL, and …”. Please, change for “synthesis whereas the other phenotypic parameters (TC, TG, LDL-c, and…’

            Please use HDL-c instead of HDL-C (in the Abstract and along with the text).

            I suggest including “mice” as a keyword.

INTRODUCTION

            I suggest that authors mention the term MAFLD (Metabolic associated fatty liver disease (MAFLD) since that “there is substantial evidence showing the superior use of the MAFLD definition over that of NAFLD”. As references, please see the following good examples: 10.1016/S2468-1253(22)00062-0 //10.12998/wjcc.v10.i20.6759 // 10.1152/ajpcell.00232.2022

            Please, remove the underline from lines 55-57.

            References 4,5, and 6 are cited in a different font or font size.

            Please, include references of 2022 in this section. There are so many in PUBMED (as the DOI I showed before).

METHODS

            This section is well prepared. Please, change “2.4.3 HPLC-Mass spectrometry” for “2.4.3 HPLC-Mass spectrometry” (Line 134).

            Please, mention the weight of the animals.

RESULTS

            This section is very rich. Figures are very good.

            Please, change “Figure 6.. The…” for “Figure 6. The

            The authors mention, for example, Figure 3 A, Figure 4 A, B and so on… However, the letters “A”, “B”… do not appear in the Figures.

            In line 416, please change “(Fig 7 A, B.C)” for “(Fig 7 A, B,C), P”.

            In line 444 we can see “FAS,, BAX, ICAM1, OCLN, GSTP1,…” Please, change for “FAS, BAX, ICAM1, OCLN, GSTP1,…” (remove the extra coma).

            For Figure 8 legend:

“Figure 8. The anti-inflammation property of Torularhodin. The heatmap of inflammation-related proteins (A) and schematic illustration of the possible distribution of these proteins in the cells. Nf-kB and STAT3 are transcription factors.” Should it be “Figure 8. The anti-inflammation property of Torularhodin. The heatmap of inflammation-related proteins (A) and schematic illustration of the possible distribution of these proteins in the cells. Nf-kB and STAT3 are transcription factors (B).” The letters are, again, not shown in the Figure.

The information found in lines 506-521 should be inserted in the Discussion section. The same for 527-529.

DISCUSSION

            This section is adequately performed. However, I think it would be appropriate if the authors could include the study's limitations.

CONCLUSION

            In line 588 we see “…and cholesterol execration accompanying reduced…” Is execration the correct word to be used here?

Round 2

Reviewer 1 Report

The authors responded comments. It is acceptable.